# Doubly Structured Data Synthesis for Time-Series Energy-Use Data

**DOI:** 10.3390/s24248033

**Published:** 2024-12-16

**Authors:** Jiwoo Kim, Changhoon Lee, Jehoon Jeon, Jungwoong Choi, Joseph H. T. Kim

**Affiliations:** 1Department of Statistics and Data Science, Yonsei University, 50, Yonsei-ro, Seodaemun-gu, Seoul 03722, Republic of Korea; zioo1004_@yonsei.ac.kr; 2Korea Smart Grid Institute, 3F, Samwoo Bldg, 32, Nonhyeon-ro 86 gil, Gangnam-gu, Seoul 06223, Republic of Korea; chrislee@smartgrid.or.kr (C.L.); cjungwoong@smartgrid.or.kr (J.C.); 3Department of Applied Statistics, Yonsei University, 50, Yonsei-ro, Seodaemun-gu, Seoul 03722, Republic of Korea; wjswpgns0324@yonsei.ac.kr

**Keywords:** data augmentation, energy data, energy management, electronic energy use, data privacy, synthetic data

## Abstract

As the demand for efficient energy management increases, the need for extensive, high-quality energy data becomes critical. However, privacy concerns and insufficient data volume pose significant challenges. To address these issues, data synthesis techniques are employed to augment and replace real data. This paper introduces Doubly Structured Data Synthesis (${\mathrm{DS}}^{2}$), a novel method to tackle privacy concerns in time-series energy-use data. ${\mathrm{DS}}^{2}$ synthesizes rate changes to maintain longitudinal information and uses calibration techniques to preserve the cross-sectional mean structure at each time point. Numerical analyses reveal that ${\mathrm{DS}}^{2}$ surpasses existing methods, such as Conditional Tabular GAN (CTGAN) and Transformer-based Time-Series Generative Adversarial Network (TTS-GAN), in capturing both time-series and cross-sectional characteristics. We evaluated our proposed method using metrics for data similarity, utility, and privacy. The results indicate that ${\mathrm{DS}}^{2}$ effectively retains the underlying characteristics of real datasets while ensuring adequate privacy protection. ${\mathrm{DS}}^{2}$ is a valuable tool for sharing and utilizing energy data, significantly enhancing energy demand prediction and management.

## 1. Introduction

In recent years, there has been an increasing interest in utilizing and modeling energy-related large datasets. For example, to improve the accuracy of time-series data prediction in smart grids, researchers have introduced advanced short-term load prediction methods using matrix factorization [1] and federated machine-learning frameworks [2]. Efforts are being made to find effective solutions for classifying and clustering time-series data to solve real-world problems, such as detecting building occupancy, energy theft, and cyber attacks [3,4,5,6,7,8,9,10]. In addition, research has been conducted on missing value imputation [11] and privacy-preserving methodologies [12], which have improved the reliability and efficiency of smart grids.

Despite the growing demand for data application within the energy sector, a critical challenge lies in addressing data privacy concerns. A major issue regarding the privacy issue of energy datasets is that they often contain personal or household privacy information. For example, the dataset may reveal a household’s daily life and habits through electricity usage patterns and activity times. The privacy concern is a pertinent barrier in energy research and applications as privacy-related regulations often limit further data analyses, sharing, and publication [13]. To address these challenges, the deployment of synthetic data has become increasingly essential. Synthetic data of energy data enables the sharing and disclosure of privacy-protected data. It is no longer necessary to disclose original data that contain sensitive privacy information.

Synthetic data are data that have been generated using a purpose-built mathematical model or algorithm, with the aim of solving data science tasks [14]. As artificially generated datasets, they minimize the risk of re-identification of the individual. At the same time, by carefully designing the synthesis algorithm, a synthetic dataset can preserve the statistical structure and properties of the original dataset, allowing for various quantitative analyses, of which the results are practically the same as the analyses carried out on the original dataset. In other words, the use of synthetic data strikes a balance between data utility and privacy by anonymizing micro-level data while maintaining the usefulness of energy data. However, existing approaches, particularly those based on Generative Adversarial Networks (GANs) [15,16], often fail to fully capture the time-series nature of energy datasets, limiting their effectiveness in preserving temporal dependencies.

This paper addresses these challenges by proposing a general time-series energy-use data synthesis method designed to preserve both the longitudinal and cross-sectional characteristics of energy datasets. Our approach introduces two key innovations. First, to preserve the longitudinal information we synthesize the rate changes in energy consumption instead of the original consumption values. This conversion turns out to be effective as it can naturally link two consecutive values in a time-series dataset. Second, we apply a calibration technique so that the synthetic dataset has the same mean as the original datasets at each time point. This way the cross-sectional property of the original dataset can be preserved. Our numerical analysis using a real data from a condominium shows that the new method works better than the existing alternative synthesis methods. The contributions of this study are demonstrated through numerical analyses using real data from a condominium, showing that the proposed method outperforms existing alternatives in reproducing the time-series nature of energy datasets.

This paper is organized as follows: Section 2 reviews previous work on synthetic data generation for energy data. Section 3 introduces the methodology proposed in this study. Section 4 includes an introduction to the target energy usage data, followed by the presentation of evaluation metrics and results. Finally, Section 5 concludes with a discussion of limitations and outlines future research directions.

## 2. Related Work

The synthesis of time-series energy use has been conducted mainly around GANs. For example, Conditional Tabular GAN (CTGAN) [17] has been popularly used for data synthesis. The model employs a Conditional GAN to model probability distributions, incorporating a mode-specific normalization approach and training-by-sampling technique. This approach effectively handles multimodal distributions in continuous variables and imbalanced distributions in discrete variables. However, CTGAN often fails to capture original data structure because it is not originally designed for time-series data.

To capture the temporal dependence of time-series data and achieve stable learning, Fekri et al. [18] introduced R-GAN (Recurrent-GAN), an application of Wasserstein GAN (WGAN) [19] and Metropolis–Hastings GAN (MH-GAN) [20]. The R-GAN consists of two stacked Long Short-Term Memorys (LSTMs) [21] and uses the WGAN algorithm, which updates the discriminator multiple times before updating the generator. The WGAN algorithm is used to mitigate the problem of mode collapse, where either the Generator or the Discriminator become too well trained. However, if it falls into mode collapse, it does not generate diverse data and converges to a certain mode. To avoid this, the MH-GAN method is adopted, which ensures diversity and maintains the quality of the generated data by having the discriminator pick and choose the data most similar to the actual data among the data generated by the Generator. In this way, R-GAN combines WGAN and MH-GAN to capture temporal dependence in time-series data and generate stable and diverse data at the same time.

Also, Li et al. [22] proposed a Transformer-based Time-Series Generative Adversarial Network (TTS-GAN) designed to handle long-length time-series data, often seen in medical machine learning, where datasets are small and relationships are irregular. TTS-GAN uses transformer-based attention mechanisms, which compare each value to many others, to overcome the gradient loss issues typical of Recurrent Neural Networks (RNNs). This attention allows the model to consider multiple relevant values from different points in the time series simultaneously, ensuring important past values influence the current value. To manage long time-series data, positional embedding is employed, assigning unique vectors to represent each value’s position, ensuring contextualization within the series. TTS-GAN proves effective in generating long, multidimensional time-series energy-use data, addressing the challenges posed by extensive data and small datasets. Table 1 summarizes the key features, advantages, and limitations of these models.

## 3. Methods

In this section, we introduce a new data synthesis method called Doubly Structured Data Synthesis (${\mathrm{DS}}^{2}$). The proposed algorithm is designed to capture both cross-sectional and time-series data structures. Details of the algorithm are described step-by step in Algorithm 1. Figure 1 illustrates the overall process of the proposed methods in the diagram.
**Algorithm 1:**  ${\mathrm{DS}}^{2}$ Algorithm.
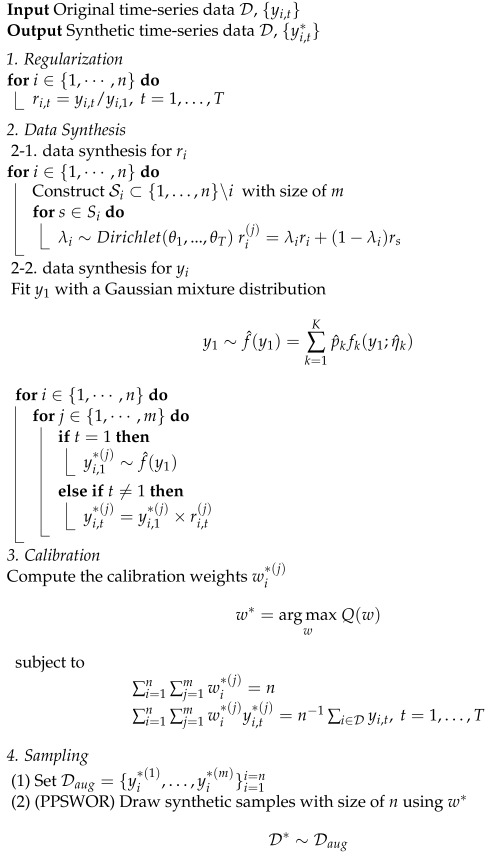


### 3.1. Regularization

Let D be the original data with a sample size of *n* and a time length of *T*. Here, $y_{i,t}$ denotes the positive outcome of unit i(=1,…,n), measured at time t (=1,…,T). To handle the time-series data structure, we convert the target outcomes to scaled values with $r_{i,t}=y_{i,t}/y_{i,1}$, where $r_{i,t}$ are the relative changes from the baseline observation $y_{i,1}>0$. Furthermore, we define $r_{i}=\left(r_{i,1},\ldots ,r_{i,T}\right)$ as the vectorized expression of the relative changes. This regularization helps to preserve the time-series structure in the sense that the synthetic values are constructed using the relative changes rather than the changes in original scale. This step is more efficient for large *T*.

### 3.2. Data Synthesis

Data synthesis involves two sub-procedures: (1) generating relative changes and (2) creating the synthetic outcomes. To avoid identity disclosure risks, we first create new relative changes using interpolation. For each unit *i*, *m* candidates of the relative changes are generated in a mixed-up way,
$$r_{i}^{\left(j\right)}=\lambda_{i}r_{i}+\left(1-\lambda_{i}\right)r_{j},$$
where $r_{j}$, j=1,…,m, are randomly chosen from the other actual relative changes. The vector of mixing ratio $\lambda_{i}$ is here generated from a Dirichlet distribution with $\left(\theta_{1},\ldots ,\theta_{T}\right)$. An important characteristic of the Dirichlet distribution is that the generated values are between 0 and 1 and their sum is equal to 1. The nuisance parameters θ in the Dirichlet distribution controls the size of variations in each time window. The large values with $\theta_{1}=\cdots =\theta_{T}$ produce similar values in the range of $\lambda_{i,1},\cdots ,\lambda_{i,T}$. If the disclosure risk is not a critical issue, each mixing ratio can be independently generated from a uniform distribution, for example,
$$\lambda_{i,t}∼\mathrm{Uniform}\left(\theta_{t},1\right)$$
with a large enough $0.5\leq \theta_{t}<1$.

After the synthetic relative changes are generated, we generate synthetic values for the target outcomes with a two-step procedure:(1)Generate synthetic values for $y_{i,1}^{\left(j\right)}$ from a fitted Gaussian mixture distribution.(2)Produce synthetic values for $y_{i,t}^{\left(j\right)}$ with
$$y_{i,t}^{*(j)}=y_{i,1}^{*(j)}\times r_{i,t}^{\left(j\right)}.$$

To generate the starting value, we first fit $y_{i,1}$ into a Gaussian Mixture Distribution (GMM),
$$f\left(y_{i,1};p,\eta \right)=\sum_{k=1}^{K}p_{k}f_{k}\left(y_{i,1};\eta_{k}\right),$$
where $p=(p_{1},\ldots ,p_{K})$ and $f_{k}$ is a Gaussian distribution with $\eta =(\eta_{1},\ldots ,\eta_{K})$. The GMM parameters are estimated using the conventional EM algorithm [23]. This estimation procedure can be replaced with other parametric, semi-parametric, and non-parametric techniques.

### 3.3. Calibration

As the results of data synthesis, we have n×m observations. However, those observations may not preserve the cross-sectional data structure because preserving time structure is only considered during the data synthetic procedure. To capture the cross-sectional data structure, we apply a calibration technique [24,25] so that the calibration weights $w_{i}^{*(j)}$ satisfy the following two constraints: (1)$$\begin{array}{ccc} \sum_{i=1}^{n}\sum_{j=1}^{m}w_{i}^{*(j)} & = & n, \end{array}$$(2)$$\begin{array}{ccc} \sum_{i=1}^{n}\sum_{j=1}^{m}w_{i}^{*(j)}y_{i,t}^{*(j)} & = & n^{-1}\sum_{i\in D}y_{i,t},\;t=1,\ldots ,T. \end{array}$$

Constraint (Equation 1) implies that we treat the synthetic samples with a size of *n* even if their actual size is n×m. Also, constraint (Equation 2) indicates that the weighted mean of synthetic values should be similar to the value of the original values in each time window. Furthermore, if a specific condition should be preserved between the original and synthetic datasets, we can add further conditions with any continuous function $g(y_{i,t})$, such that
$$\sum_{i=1}^{n}\sum_{j=1}^{m}w_{i}^{*(j)}g\left(y_{i,t}^{*(j)}\right)=n^{-1}\sum_{i\in D}g\left(y_{i,t}\right)$$

To find the calibration weights, we use a Lagrange method [25] with a loss function in Equation (Equation 3),
(3)$$Q\left(w\right)=\sum_{i=1}^{n}\sum_{j=1}^{m}w_{i}^{\left(j\right)}\log w_{i}^{\left(j\right)}/\pi_{i}^{\left(j\right)},$$
where $\pi_{i}^{\left(j\right)}$ are the initial weights assigned on unit *i*. Combining the loss function and constraints, we can construct a Lagrangian function,
(4)$$\begin{array}{ccc} L(w) & = & \sum_{i}\sum_{j}w_{i}^{\left(j\right)}\log w_{i}^{\left(j\right)}/\pi_{i}+\lambda_{1}\left(\sum_{i}\sum_{j}^{m}w_{i}^{*(j)}-n\right)+\lambda_{2}\left(\sum_{i}\sum_{j}w_{i}^{*(j)}y_{i,t}^{*(j)}-{\bar{y}}_{t}\right) \end{array}$$

The optimal values of $w_{i}^{*(j)}$, which minimizes the Lagrangian function (Equation 4), can be expressed as
(5)$$w_{i}^{*(j)}\left(\lambda \right)\propto \pi_{i}^{\left(j\right)}\exp\left(-\lambda_{1}-\lambda_{2}y_{i,t}^{*(j)}\right),$$
and so we can rewrite constraints (Equation 1) and (Equation 2), incorporating the results in (Equation 5),
(6)$$\begin{array}{ccc} \sum_{i=1}^{n}\sum_{j=1}^{m}\pi_{i}^{\left(j\right)}\exp\left(-\lambda_{1}-\lambda_{2}y_{i,t}^{*(j)}\right) & = & n, \end{array}$$


(7)
$$\begin{array}{ccc} \sum_{i=1}^{n}\sum_{j=1}^{m}\pi_{i}^{\left(j\right)}\exp\left(-\lambda_{1}-\lambda_{2}y_{i,t}^{*(j)}\right)y_{i,t}^{*(j)} & = & {\bar{y}}_{t}. \end{array}$$


From these results, we can compute the calibration weights by solving the estimating Equations (Equation 6) and (Equation 7) with conventional optimization techniques, such as Newton–Raphson or gradient descent algorithms. Note that the calibration weights should be updated several times across *T* time windows. See [25] for more details.

### 3.4. Sampling (PPSWOR)

We now have n×m synthetic values, which preserve both time-series and cross-sectional data structures. From these initial candidates, we select the final synthetic samples with a target sample size $n^{*}\ll n\times m$. In this study, we assume $n^{*}=n$ for simplicity. To capture the original data structure, we apply a Probability-Proportional-to-Size Without Replacement (PPSWOR) sampling technique, widely-utilized in survey sampling [26]. This technique ensures that the selection probability of each unit is directly proportional to its size or weight within the target population. The detailed procedures are summarized below:(Step 1)Normalize the calibration weights,
$$p_{i}^{*(j)}=\frac{w_{i}^{*(j)}}{\sum_{i}\sum_{j}w_{i}^{*(j)}}$$(Step 2)Enumerate the normalized calibration weights and n×m synthetic time-series vectors in order of s=1,…,n×m;(Step 3)Generate a random number, *u*, from a standard uniform distribution;(Step 4)Select the *s*-th synthetic time-series vector as a final sample,
$$s=\underset{s}{\arg\min}\left(C_{s}-u\right),$$
where
$$C_{s}=\sum_{v=1}^{s}p_{v}^{*}$$(Step 5)Repeat Step 3 and Step 4 without replacement until $n^{*}$ units have been selected.

Since without replacement sampling is applied in this design, the number of replicates, *m*, should be large enough to preserve the original data structure. Note that m=5 or m=10 is popularly used in practice.

## 4. Experiments and Results

### 4.1. Summary of Data

To check the performance of our proposed algorithm, we applied it to the synthesis of the Advanced Metering Infrastructure (AMI) data. AMI is an integrated system of smart meters, communication networks, and data management systems that can remotely measures consumption data in real time for utilities such as gas, electricity, water, and heat. Thus, AMI plays a pivotal role in providing detailed and real-time insights into energy usage patterns.

The AMI data in this research, collected by the Korea Smart Grid Institute, encompass the records of energy consumption observed in 9986 individual households over nine condominiums in South Korea, spanning April to July 2022. The dataset includes condominium name, household IDs, and monthly electricity consumption. See Table 2 for the average monthly electricity consumption in the nine condominiums. Note that the increasing trend in consumption suggests seasonality, influenced by rising temperatures and greater use of cooling appliances like air conditioners. Also, each household has different energy consumption patterns due to the different number of household members and life cycles.

### 4.2. Evaluation Metrics

We consider three data synthesis methods: ${\mathrm{DS}}^{2}$(Ours), CTGAN, and TTS-GAN. Also, we consider three types of metric, including data similarity, data utility, and data privacy, to evaluate different aspects of the synthetic data. Details on each part are outlined below.

#### 4.2.1. Data Similarity

This part evaluates statistical similarities and the comparison of descriptive statistics at the distributional level. In particular, we use the following specific metrics in this part.

Statistical Comparison and Visualization: To gauge the statistical similarity, the original and a synthetic datasets are compared in terms of standard descriptive statistics, such as mean value (μ) and standard deviation (σ); clearly, close values indicate high similarity between the two datasets. In addition, kernel density estimation results are presented for a visual comparison of distributional similarities. The degree of overlap between two kernel densities visually indicates the similarity between the probability distributions.Average Wasserstein Distance (Avg WD): The Wasserstein distance represents the minimum cost between two probability distributions and is calculated via $W\left(P,Q\right)=\inf_{\gamma \in \prod (P,Q)}\int_{X\times Y}c\left(x,y\right)\;d\gamma \left(x,y\right)$, where γ is the joint probability distribution in the probability measure space, ∏(P,Q), and c(x,y) is the cost function. A smaller value of Avg WD indicates that the two datasets are close to one another, while a larger value suggests that the two datasets are far away from one another.Difference in Correlation (Diff. Corr.): This measures the difference between the correlation of the two datasets. It is calculated by $Diff.Corr.=|\rho_{synthetic}-\rho_{original}|$, where the correlation coefficient is obtained by $\rho \left(X,Y\right)=Cov\left(X,Y\right)/\left(\sigma_{x}\sigma_{y}\right)$, where Cov(.,.) is the covariance and $\sigma_{x}$ and $\sigma_{y}$ are the respective standard deviations. A smaller Diff.Corr. value means that the two datasets are similarly correlated, while a larger value indicates that the correlation between the two datasets is considerably different.

#### 4.2.2. Data Utility

Data utility represents the usefulness of the synthetic data, indicating the extent to which the synthetic data can be used for a specific analysis or model training. In this research, we evaluate how similar the electricity bills generated using the synthetic dataset are to those generated using the original dataset. The smaller the difference between the two amounts, the more representative the synthetic data is of the original data and the more likely it is to provide actionable information.

In South Korea, electricity bills are composed of the sum of the basic charge, electricity volume charge, climate environment charge, and fuel cost adjustment charge. Table 3 presents the calculation details for both the base rate and metered rate. More specifically:Climate environment charge = Climate environment charge unit price × electricity consumptionFuel cost adjustment charge = Fuel cost adjustment unit price × electricity consumption

The total electricity bill amount is calculated as the sum of the electricity bill, VAT, and electricity industry base fund.

Value-Added Tax (VAT): Electricity bill × 10%.Electricity industry base fund: Electricity bill × 3.7%

**Table 3 sensors-24-08033-t003:** Base rate and metered rate calculation table.

Base Rate (KRW/Unit)		Metered Rate (KRW/kWh)	
200 kWh or less used	730	First 200 kWh	78.2
201–400 kWh used	1260	Next 200 kWh used	147.2
More than 400 kWh used	6060	Over 400 kWh	215.5

#### 4.2.3. Data Privacy

Data privacy measures how well the synthetic data protects the privacy of the original data provider. If it is difficult to identify or extract sensitive information from the synthetic dataset, the synthetic dataset is deemed highly protective. The data privacy metrics used in this research are presented below.

Distance to Closest Record (DCR): DCR evaluates the ability to identify an individual by calculating the Euclidean distance of the closest record between the synthetic dataset and the original one. The higher the DCR, the lower the likelihood of identification risk in the synthetic data. DCR is defined as
$$DCR=\frac{1}{n}\sum_{i=1}^{n}min\_distance\left(record_{i}\right)$$Nearest Neighbor Distance Ratio (NNDR): NNDR represents the ratio of the Euclidean distance between the nearest and next closest neighbors of each synthetic data to the original data, which is in the range [0, 1]. Higher NNDR values indicate better privacy, while lower values indicate the possibility of leaking sensitive information from the nearest original data.
$$NNDR=\frac{min\_distance(record_{i})}{second\_min\_distance(record_{i})}$$Data Utility and Privacy Index (DUPI): DUPI [27] was originally designed to evaluate both data similarity and privacy simultaneously. Given the trade-off between data similarity and data privacy, we use this metric for the data privacy evaluation. DUPI and optimal score, $E_{H_{0}}\left(DUPI\right)$, is calculated using the following equations:
$$DUPI=\frac{1}{n_{ori}}\sum_{i=1}^{n_{ori}}I\left[min\left(d\left(x_{i},x_{j}\right):j\neq i\right)\geq min\left(d\left(x_{i},y_{j}\right):j\right)\right]$$
$$E_{H_{0}}\left(DUPI\right)=\frac{n_{syn}}{n_{ori}+n_{syn}-1},$$
where $x_{i}$ and $y_{j}$ are the *i*-th and *j*-th observations of the original and synthetic datasets, with sizes of $n_{ori}$ and $n_{syn}$, respectively. Here, *d* can be any metric (distance) function. The value of DUPI can range from 0 to 1, and its expected value becomes approximately 0.5 when $n_{ori}=n_{syn}$, which represents the optimal case. For a given synthetic dataset, a larger DUPI means higher utility and lower privacy protection, and a smaller DUPI suggests lower utility and higher privacy protection. Addressing the utility–privacy trade-off is essential for synthetic data generation, particularly when sensitive information is involved, as is the case for energy data, which often contain private information. The use of the DUPI metric is especially relevant in this context, as it provides a comprehensive assessment by identifying the optimal balance between data utility and privacy, offering a more effective evaluation compared to conventional metrics. DUPI can be calculated for a given synthetic energy dataset to evaluate the balance between utility and privacy. A DUPI value greater than 0.5 indicates that the synthetic data exhibits higher utility but lower privacy protection, whereas a value less than 0.5 suggests lower utility and higher privacy protection. A DUPI value equal to or close to 0.5 signifies that the synthesis achieves an optimal balance between utility and privacy, aligning with the characteristics of the original energy dataset. Later we will present a graphical representation of the results obtained by applying DUPI to the dataset.

### 4.3. Results

The data synthesis results are presented below. For simplicity, we have summarized the results for three condominiums: Gongdeok A (Condominium 1), Daegu G (Condominium 2), and Dongjak I (Condominium 3). The results for other condominiums are similar to those of these selected three.

#### 4.3.1. Data Similarity

The comparison of mean and standard deviations of the average monthly electricity energy use for the three selected condominiums is presented in Table 4. ${\mathrm{DS}}^{2}$ demonstrates the closest alignment to the original data in both metrics, outperforming CTGAN and TTS-GAN in nearly all cases. This highlights the superior ability of ${\mathrm{DS}}^{2}$ to accurately replicate the central tendencies and variability of the original dataset.

Figure 2 presents the estimated kernel density functions constructed from households’ original and synthetic electricity usages. TTS-GAN demonstrates notable deviations from the original data across all scenarios, failing to replicate the distribution accurately. On the other hand, CTGAN captures the overall shape reasonably well but tends to overemphasize peaks in certain cases. In contrast, ${\mathrm{DS}}^{2}$ achieves the closest alignment with the density of the original dataset.

Also, Figure 3 shows the monthly electricity usage in kWh for Condominium 1 against time in months. It is evident that the synthetic dataset produced from the ${\mathrm{DS}}^{2}$ method is most similar to the original dataset, capturing the non-decreasing trend over the observation time period. Also, the other two methods exhibit unnatural fluctuations over time, which are not found in the original dataset and the synthetic one from the ${\mathrm{DS}}^{2}$ method.

As the final similarity comparison, the Avg WD and Dff. Corr. are computed and compared in Table 5. For the Avg WD side, we see that both ${\mathrm{DS}}^{2}$ and CTGAN have small distribution differences from the original dataset for all three condominiums, indicating that the synthetic dataset reflects the original one well. On the other hand, the Diff. Corr. comparison shows that both CTGAN and TTS-GAN yield quite large values, implying significant deviation from the original dataset. This shows that GAN-based synthesis methods are rather limited in their ability to model the dependence structure of the original dataset.

#### 4.3.2. Data Utility

Turning to data utility, Table 6 compares three synthesis methods with estimated electricity bills converted from synthetic electricity usages. We see that, for all three condominiums, the performance of the ${\mathrm{DS}}^{2}$ method is the best in terms of similarity with the original bills.

The distribution-wise visual comparison based on kernel density estimation for the electricity bill amounts is given in Figure 4. The figures show that ${\mathrm{DS}}^{2}$ create a synthetic dataset that is most similar to the original one in terms of the density shape. CTGAN is also capable of producing good synthetic datasets but is sometimes inadequate in properly describing the peak area in the density. Unfortunately, TTS-GAN is subpar in most cases, as shown in the figure. In fact, this visual comparison gives us the same conclusion we have drawn from the data similarity counterparts in Figure 2.

#### 4.3.3. Data Privacy

The data privacy is evaluated using the DCR and NNDR metrics introduced earlier. The results in Table 7 show that ${\mathrm{DS}}^{2}$ has relatively low DCR and NNDR values compared to CTGAN and TTS-GAN, suggesting that it is a model that balances privacy and utility.

The DUPI results are provided in Table 8. As we set the size of the synthetic datasets to be the same as that of the original dataset, the optimal DUPI value is close to 0.5. From the values in Table 8 we can see that, for all three condominiums, the ${\mathrm{DS}}^{2}$ method produces relatively stable DUPI values close to the optimal value, confirming the effectiveness of the method in maintaining the utility and privacy of the original data. The other two methods, CTGAN and TTS-GAN, give much smaller DUPI values close to zero, which implies that these GAN-based methods tend to achieve a high level of privacy protection in exchange for substantial loss in data utility. The results presented in Table 8 are illustrated in Figure 5. The x-axis represents the utility metric, while the y-axis represents the privacy metric, with both ranging from 0 to 1. Along the pink curve, the DUPI values are derived, reflecting the trade-off relationship between utility and privacy. In Figure 5, the intersection of the gray dotted lines represents the point where utility and privacy are balanced. It can be observed that the result of ${\mathrm{DS}}^{2}$ is the closest to this point.

## 5. Conclusions

The generation of high-quality synthetic data that can preserve the intricate structure of the original energy-use data and keep the privacy risk minimum is an important issue in data-driven applications and policies in the energy sector.

In this paper, we have introduced a novel data synthesis approach, the ${\mathrm{DS}}^{2}$ methodology, aimed at safeguarding the privacy of AMI-based power consumption data while retaining its essential cross-sectional and time-series characteristics. Carefully crafted with the Dirichlet distribution and Gaussian Mixture models, and further with weight calibration and PPS sampling techniques, this innovative method has solid theoretical foundations, with a great potential to enhance the utility of energy data sharing and analysis.

The simulation results, employing real AMI data, show the exceptional performance of the ${\mathrm{DS}}^{2}$ method. Various comparative analyses demonstrate that the ${\mathrm{DS}}^{2}$ method outperforms existing synthesis methods such as CTGAN and TTS-GAN by accurately preserving the underlying characteristics of the actual AMI dataset with an adequate level of privacy protection. In contrast, CTGAN fails to capture temporal dependencies effectively, limiting its applicability to time-series energy data. TTS-GAN, while capable of handling long time series, requires a significant number of long-term time points for effective training. These limitations highlight the robustness and adaptability of the ${\mathrm{DS}}^{2}$ method in diverse energy data synthesis scenarios.

The proposed method pioneers data synthesis by integrating both the time-series and cross-sectional characteristics of energy consumption data, yet there remains room for future research. First, handling extreme values is a critical challenge in data synthesis due to their rarity and the risks they pose. Retaining extreme values without proper treatment can heighten privacy risks, while eliminating them may distort analyses and predictions for the quantity of interest. A more balanced approach involves implementing robust outlier detection mechanisms to manage extreme values effectively while preserving the utility of the synthesized data. To address this, we are currently enhancing the ${\mathrm{DS}}^{2}$ method by developing a novel utility–privacy scoring framework at the record level, enabling controlled treatment of extreme observations based on their utility–privacy trade-offs.

Future research will also explore the scalability and robustness of ${\mathrm{DS}}^{2}$ when applied to larger datasets or longer time series to ensure its utility and privacy-preserving capabilities. Furthermore, efforts will focus on advancing privacy-preserving algorithms and expanding the applicability of ${\mathrm{DS}}^{2}$ to a wider range of domains, enhancing its adaptability to diverse energy data scenarios. These advancements in synthetic data techniques will significantly improve the sharing and utilization of energy data, providing vital support for energy demand prediction and management.

## Figures and Tables

**Figure 1 sensors-24-08033-f001:**
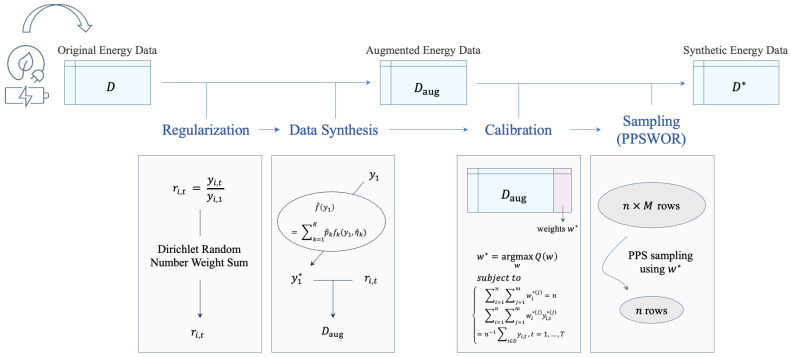
Diagram illustrating the overall process of the proposed methods.

**Figure 2 sensors-24-08033-f002:**
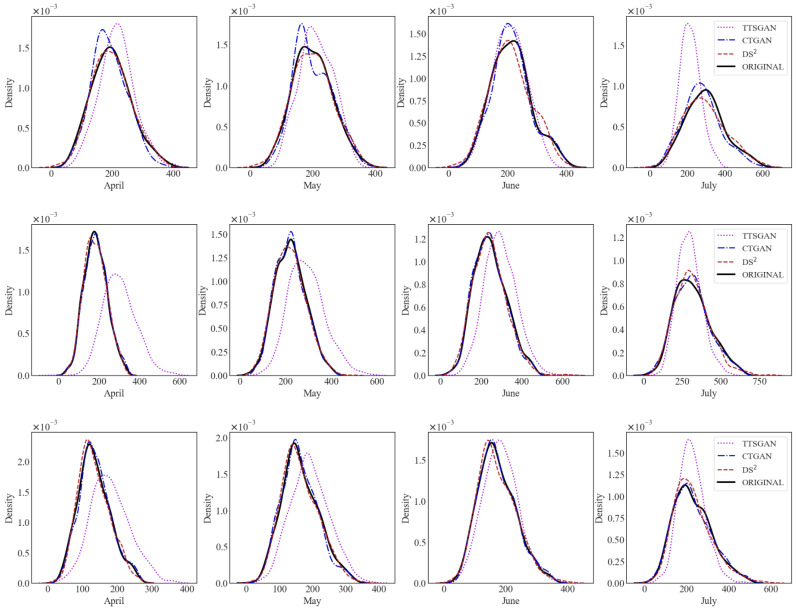
Illustrative examples of density similarity in monthly households’ electricity use (**Top**: Condominium 1, **Middle**: Condominium 2, **Bottom**: Condominium 3).

**Figure 3 sensors-24-08033-f003:**
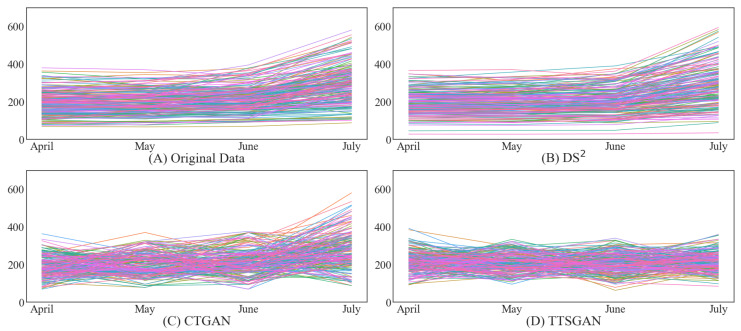
Data similarity: monthly electricity usage in kWh for Condominium 1.

**Figure 4 sensors-24-08033-f004:**
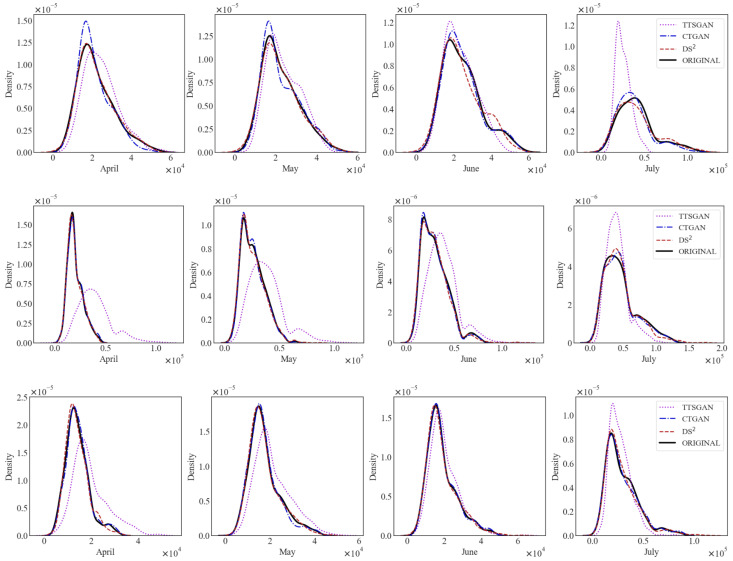
Illustrative examples of density similarity in monthly households’ electricity bills (**Top**: Condominium 1, **Middle**: Condominium 2, **Bottom**: Condominium 3).

**Figure 5 sensors-24-08033-f005:**
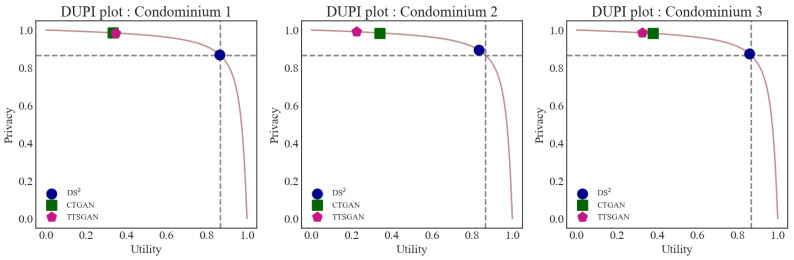
DUPI plot (**Left**: Condominium 1, **Middle**: Condominium 2, **Right**: Condominium 3).

**Table 1 sensors-24-08033-t001:** Model comparison for time-series energy data synthesis.

Model	Key Techniques	Advantages	Limitations
CTGAN	Uses conditional GAN, mode-specific normalization, and training-by-sampling	Handles multimodal distributions, balances discrete variables	Fails to capture temporal dependencies
R-GAN	Uses WGAN and MH-GAN	Captures temporal dependencies, mitigates mode collapse, ensures diversity	Requires complex model structure and careful tuning
TTS-GAN	Applies transformer-based attention and positional embedding	Works effectively for long time-series, handles small datasets, retains relationships	Demands high computational resources

**Table 2 sensors-24-08033-t002:** Condominium information and average monthly electricity consumption (kWh).

	Condominium			Average Monthly Electricity Consumption (kWh)			
**No.**	**Name**	**Location**	**# of Units**	**April**	**May**	**June**	**July**
1	Gongdeok A	Seoul	184	197.94	200.50	214.75	296.47
2	Osan B	Osan	1204	190.13	191.22	213.79	296.58
3	Dongtan C	Suwon	1836	138.68	141.57	154.78	207.33
4	Hwaseong D	Hwaseong	1173	149.45	181.01	197.65	276.70
5	Dongjak E	Seoul	156	135.08	161.18	168.49	221.30
6	Songpa F	Seoul	1976	167.69	204.08	236.27	343.55
7	Daegu G	Daegu	1857	183.45	222.37	245.98	318.25
8	Mapo H	Seoul	767	158.40	192.82	218.45	331.88
9	Dongjak I	Seoul	833	134.62	163.96	173.28	229.21

**Table 4 sensors-24-08033-t004:** Data similarity evaluation.

		April		May		June		July	
		**Mean**	**Std.**	**Mean**	**Std.**	**Mean**	**Std.**	**Mean**	**Std.**
Condominium 1	Original	197.94	63.48	200.50	61.17	214.75	66.75	296.47	103.53
	${\mathrm{DS}}^{2}$	198.39	63.54	200.28	64.39	211.14	69.03	293.34	111.76
	CTGAN	191.16	56.85	201.07	60.20	214.51	65.96	278.04	98.40
	TTS-GAN	215.96	56.22	211.76	52.60	207.50	55.93	215.33	52.91
Condominium 2	Original	183.45	56.67	222.37	67.53	245.98	80.14	318.25	116.17
	${\mathrm{DS}}^{2}$	182.27	56.38	220.14	69.66	243.19	82.26	315.25	114.36
	CTGAN	185.99	58.25	221.33	66.14	242.27	78.91	314.17	115.42
	TTS-GAN	300.81	82.16	292.51	79.77	294.52	79.18	297.73	79.75
Condominium 3	Original	134.62	45.38	163.96	55.49	173.28	59.85	229.21	88.67
	${\mathrm{DS}}^{2}$	133.44	44.52	162.52	55.31	172.51	61.70	229.30	89.98
	CTGAN	137.22	45.35	161.63	54.13	174.41	59.93	229.22	91.19
	TTS-GAN	179.88	55.80	192.14	57.16	184.76	57.15	224.01	61.68

**Table 5 sensors-24-08033-t005:** Data similarity evaluation: Avg WD and Diff.Corr.

		Avg WD	Diff. Corr.
Condominium 1	${\mathrm{DS}}^{2}$	0.3	0.07
	CTGAN	0.4	3.11
	TTS-GAN	0.8	3.02
Condominium 2	${\mathrm{DS}}^{2}$	0.01	0.04
	CTGAN	0.01	3.08
	TTS-GAN	0.13	3.00
Condominium 3	${\mathrm{DS}}^{2}$	0.01	0.06
	CTGAN	0.01	3.18
	TTS-GAN	0.08	3.24

**Table 6 sensors-24-08033-t006:** Comparison of mean and standard deviations of synthetic energy bills (Top: Condominium 1, Middle: Condominium 2, Bottom: Condominium 3).

	April		May		June		July	
	**Mean**	**Std.**	**Mean**	**Std.**	**Mean**	**Std.**	**Mean**	**Std.**
Original	22,246	9198	22,575	8890	24,733	10,196	41,639	21,339
${\mathrm{DS}}^{2}$	22,327	9170	22,643	9290	24,285	10,313	41,576	23,134
CTGAN	21,158	8032	22,672	8883	24,599	10,084	38,029	19,556
TTS-GAN	24,597	8610	23,927	7928	23,418	8217	25,619	8494
Original	20,110	7687	25,944	10,448	30,101	13,751	46,583	25,211
${\mathrm{DS}}^{2}$	19,968	7610	25,676	10,908	29,743	14,307	45,770	24,785
CTGAN	20,481	7982	25,729	10,274	29,469	13,430	45,703	24,645
TTS-GAN	39,858	16,434	38,221	15,761	38,518	15,435	40,864	16,273
Original	14,135	4967	17,683	7069	18,949	8008	29,160	15,540
${\mathrm{DS}}^{2}$	14,007	4845	17,519	7038	18,916	8342	29,187	16,337
CTGAN	14,424	5044	17,343	6832	19,099	8102	29,239	16,062
TTS-GAN	19,619	7562	21,266	8049	20,283	7944	27,232	10,233

**Table 7 sensors-24-08033-t007:** Data privacy evaluation: O&S: between original and synthetic data; O: within original data, S: synthetic data.

		DCR			NNDR		
		**O&S**	**O**	**S**	**O&S**	**O**	**S**
Condominium 1	${\mathrm{DS}}^{2}$	0.191	0.158	0.212	0.400	0.437	0.393
	CTGAN	0.541	0.158	0.542	0.546	0.437	0.484
	TTSGAN	0.495	0.158	0.567	0.527	0.437	0.512
Condominium 2	${\mathrm{DS}}^{2}$	0.09	0.11	0.10	0.45	0.50	0.45
	CTGAN	0.26	0.11	0.28	0.50	0.50	0.48
	TTSGAN	0.25	0.11	0.29	0.48	0.50	0.48
Condominium 3	${\mathrm{DS}}^{2}$	0.11	0.13	0.11	0.42	0.48	0.45
	CTGAN	0.32	0.13	0.36	0.50	0.48	0.49
	TTSGAN	0.33	0.13	0.36	0.50	0.48	0.49

**Table 8 sensors-24-08033-t008:** DUPI results.

		DUPI
Condominium 1	${\mathrm{DS}}^{2}$	0.49
	CTGAN	0.10
	TTSGAN	0.10
Condominium 2	${\mathrm{DS}}^{2}$	0.44
	CTGAN	0.10
	TTSGAN	0.06
Condominium 3	${\mathrm{DS}}^{2}$	0.48
	CTGAN	0.12
	TTSGAN	0.10

## Data Availability

The datasets used in this study are available from the corresponding author upon reasonable request.

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
