# Peer review of "Doubly Structured Data Synthesis for Time-Series Energy-Use Data"

_sensors, 2024, doi:10.3390/s24248033_

Round 1
Reviewer 1 Report
Comments and Suggestions for Authors
-
The overall design of the article is clear, utilizing innovative data synthesis technology to address privacy issues while maintaining the utility of the data. The paper demonstrates the advantages of the new method by comparing various generative models.
-
Overall, the English grammar of the manuscript is good, with clear expression. However, there are some issues: Original sentence: "Synthetic data is data that has been generated using a purpose-built mathematical model or algorithm, with the aim of solving data science tasks." Issue: Subject-verb agreement error. The subject "Synthetic data" is plural, but the verb "is" is singular; Original sentence: "At the same time, by carefully designing the synthesis algorithm, a synthetic dataset can preserve the statistical structure and properties of the original dataset, allowing for various quantitative analyses of which the results are practically the same as the analyses carried out on the original dataset." Issue: "are practically same" lacks an article. Original sentence: "The constraint (1) implies that we treat the synthetic samples with a size of n even if its actual size is n × m." Issue: "its" incorrectly refers to "samples," and should be changed to "their."
-
The paper includes derivations and tests throughout; it is suggested to add an overall diagram to describe the application of the algorithm in real energy management scenarios, or to use images to describe the relationships between modules.
-
In the "related work" section, it is recommended to use a table to compare the effects of different models.
-
The research direction of the paper is on energy usage data. Does the proposed algorithm perform better on energy usage data? Please respond.
Reviewer 2 Report
Comments and Suggestions for Authors
Please see the attached file.

Round 2
Reviewer 2 Report
Comments and Suggestions for Authors
The authors significantly improved the work and addressed all the reviewers' comments. I have no more comments and I recommend the work for publication.